# Prevalence of Low Energy Availability in Competitively Trained Male Endurance Athletes

**DOI:** 10.3390/medicina55100665

**Published:** 2019-10-01

**Authors:** Amy R. Lane, Anthony C. Hackney, Abbie Smith-Ryan, Kristen Kucera, Johna Registar-Mihalik, Kristin Ondrak

**Affiliations:** 1Curriculum of Human Movement Science, University of North Carolina at Chapel Hill, Chapel Hill, NC 27516, USA; amylane18@gmail.com (A.R.L.); abbsmith@email.unc.edu (A.S.-R.); kkucera@email.unc.edu (K.K.); johnakay@email.unc.edu (J.R.-M.); 2Department of Exercise and Sport Science, University of North Carolina at Chapel Hill, Chapel Hill, NC 27516, USA; kondrak@unc.edu; 3Department of Nutrition, University of North Carolina at Chapel Hill, Chapel Hill, NC 27516, USA

**Keywords:** relative energy deficiency in sport (RED-S), exercise, eating habits, reproductive dysfunction, sex

## Abstract

*Background and Objectives:* Relative energy deficiency in sport (RED-S) has been introduced as a broad-spectrum syndrome leading to possible dysfunction in numerous physiological systems, driven primarily by low energy availability (EA). Research in females has identified specific EA cut-points indicative of risk level for developing physiological and performance disturbances. Cut-points in males have yet to be evaluated. This study examined the prevalence of low EA in competitive (non-elite), recreationally trained (CRT) male endurance athletes. *Materials and Methods*: Subjects were 108 CRT (38.6 ± 13.8 y; 12.2 ± 5.4 h/wk training) male endurance athletes (runners, cyclists, triathletes) who completed a descriptive survey online via Qualtrics^®^ and returned 3 day diet and exercise training records. EA was calculated from returned surveys and training records. Resting metabolic rate (RMR) and lean body mass (LBM) were estimated from self-reported survey data. Prevalence of risk group was categorized based on the female cut-points: At risk (AR) ≤30 kcal/kg LBM, moderate risk (MR) = 30–45 kcal/kg LBM, or no risk (NR) ≥45 kcal/kg LBM. *Results*: In this sample, 47.2% (*n* = 51) were classified as AR, 33.3% (*n* = 36) as MR, and 19.4% (*n* = 21) as NR for low EA. Cyclists had lower EA (26.9 ± 17.4 kcal/kg LBM, *n* = 45) than runners (34.6 ± 13.3 kcal/kg LBM, *n* = 55, *p* = 0.016) and all other sport categories (39.5 ± 19.1 kcal/kg LBM, *n* = 8, *p* = 0.037). *Conclusions*: The findings indicate this sample had a high prevalence of risk for low EA, at 47.2%. Only 19.4% of participants were at no risk, meaning ~80% of participants were at some degree of risk of experiencing low EA. Cyclists were at greater risk in this cohort of low EA, although why this occurred was unclear and is in need of further investigation. Future research should address whether the current female cut-points for low EA are appropriate for use in male populations.

## 1. Introduction

Relative energy deficiency in sport (RED-S) was identified in 2014 as a broad-spectrum syndrome leading to possible dysfunction in numerous physiological systems, driven primarily by the development of low energy availability (EA) [1]. Traditionally, low EA has been associated with the female athlete triad (triad) [2,3], but the introduction of the RED-S (2014) terminology has expanded the potential breadth of impact and recognizes that males too can be affected negatively by low EA [1]. Furthermore, and importantly within sport, athletes at risk of or experiencing RED-S are more likely to have an increased risk of injury and/or decreased athletic performance when training/competing [1,2,3]. In addition to injury and physiological performance disturbances, the effects of RED-S from low EA may increase the risk for future detrimental health effects later in life, e.g., earlier onset of osteoporosis.

Low EA risk is thought to be greater in aesthetic (e.g., gymnastics), weight-sensitive (e.g., jockeys, wrestling) and endurance-based (e.g., running, cycling) sports than in team or ball-based sports (e.g., soccer, basketball) [1]. The rationale for this line of thought is due in part to: (a) eating disorders or disordered eating, which can influence energy intake (EI), can be more prevalent in aesthetic and weight-sensitive sport athletes, thereby increasing their risk for low EA; and, (b) endurance sports involve high volumes of training and as such, these athletes have tremendous daily exercise energy expenditure (EEE; high caloric cost) rates, which increases their risk for low EA [4].

Extensive research in females has identified low EA cut-points indicative of risk level for the development of physiological and performance disturbances. These cut-points are: at risk (AR) ≤30 kcal/kg lean body mass (LBM), moderate risk (MR) 30–45 kcal/kg LBM, and no risk (NR) ≥45 kcal/kg LBM [5,6]. Whether male athletes share the same risk factor cut-points is currently unknown. That is, insufficient research has been conducted to determine whether there are male-specific thresholds; nonetheless, the female cut-points have been applied in male-based studies [7,8,9,10]. Furthermore, even with an understanding of the components involved in and contributing to EA, the occurrence of low EA in male athletes is not well known, as to date there has been limited research investigating the prevalence of it in male athletes [7,8,9,10,11].

Most individuals participating in sport are not at an elite level, but tend to be more health-focused and recreationally inclined with their exercise [12,13]. To that end, a majority of the existing male-based prevalence research has investigated athletes at more professional, elite levels, leaving little known about recreationally trained men’s risk for low EA [1,12,13]. Furthermore, more attention has historically been paid to female athletes regarding low EA research, due to the serious, major negative health consequences (e.g., athletic amenorrhea or osteoporosis) associated with RED-S and the triad [4].

Therefore, the purpose of this study was to investigate the prevalence of low EA in competitive (non-elite), recreationally trained male endurance athletes using an epidemiological survey approach. The survey was distributed to competitive, recreational exercise training groups in North America who were registered members of USA Track and Field, USA Cycling, and USA Triathlon, as well as comparable sport collegiate clubs.

## 2. Materials and Methods

To investigate the prevalence of low EA by risk category (AR, MR, and NR) a cross-sectional design including an online survey with diet and exercise training logs was implemented. This study was reviewed for all methods, procedures, and recruitment techniques by the Office of Human Research Ethics at the University of North Carolina, which granted approval for its implementation (Institutional Review Board study #16-3137, approved on 19th January 2017). The purpose was explained to and informed consent was obtained from the participants as they completed the initial two pages of the survey, which also notified them of the inclusion/exclusion criteria and the total study requirements and expectations.

### 2.1. Participants

The population for this study consisted of competitive, recreationally trained male endurance athletes. Participants were recruited from active running, cycling, and triathlon clubs across the United States. Inclusion criteria included exercise training for at least 10 h per week and currently training for a specific endurance event [14]. Individuals were excluded if they were under 18 years of age, currently injured, or in a ramping phase of training (i.e., acute periods of increasing/decreasing training volume–intensity).

### 2.2. Survey

#### 2.2.1. Development

The current survey was adapted from the previously validated Community Health Activities Model Program for Seniors (CHAMPS) physical activity questionnaire with guidance from a content expert [15]. Consultation and review with the University of North Carolina Odum Institute for Research in Social Sciences was also conducted to maximize survey quality and minimize respondent burden. The survey was designed to collect descriptive information regarding physical characteristics, exercise training behaviors and history, nutritional practices, injury-induced training disruptions, and upcoming competitive events. The survey was piloted by five local male endurance athletes prior to distribution and reviewed for inconsistencies, clarity, duration, and electronic issues. Upon release, this was an open survey requiring no password to complete, and it was distributed online through Qualtrics^®^ with an anonymous weblink. At the end of the survey, participants were able to request that an individualized report be emailed to them.

#### 2.2.2. Recruitment and Response Rate

Emails were sent to 22 cycling, 116 triathlon, 158 running, and 118 club teams registered with USA Track & Field, USA Cycling, and USA Triathlon, as well as similar sports collegiate clubs across the United States (414 total). Team/club contacts were asked to send the information and link to their club membership, specifically the males. A Facebook^®^ page was also created to share the link, along with dissemination through Twitter^®^. Confirmation of distribution to club team members was rare, and subsequent emails from the researchers requesting the number of team members provided few responses. While some response was received regarding roster numbers from organizations, it is impossible to accurately extrapolate the few numbers received to the potential number of overall participates contacted. Confirmed team numbers ranged from 6 to over 400 members. The recruitment email detailed the purpose of the study and included expectations of participants and link to the survey. Participants were made aware they would need to provide a personal email to receive the diet and exercise training logs. They could either return the logs through the Qualtrics^®^ survey platform or email the principal investigator (PI) directly.

A total of 396 individuals responded, with consent provided by 285 (72.0%) participants. Of these, 76.8% (*n* = 219) of consented participants completed the survey and subsequently received the diet and training record information. Of these, 49.3% (*n* = 108) of participants who completed the survey also completed the additional study activities by returning the requested diet and training records (see the following section). Only participants completing the entire survey (all relevant questions) and returning the diet and training records were included in the analysis for this study.

#### 2.2.3. Administration and Details

The descriptive survey was open from 1 February 2017, to 1 February 2018, and could be accessed through the Qualtrics^®^ link. All participants answered a minimum of 24 questions, and adaptive questioning was implemented to minimize unrelated questions (e.g., cycling questions for runners) and decrease the respondent burden. The most questions a respondent would have encountered was 38 of the 49 total questions. The survey included 15 pages with no more than four questions per page. The average time to complete the survey was approximately 16 min. Upon completion of the survey, participants were sent the diet and exercise training record forms that had been introduced before consent.

### 2.3. Diet and Exercise Training Records

Upon completion of the survey, participants received an automatic email providing further details on how to record their three days (one weekend day and two weekdays) of dietary intake and exercise training, used by investigators to assess EI and EEE, respectively. Directions for measurement and recording food along with guidance on portion size was provided (i.e., handouts and web-links to reference sites) along with the diet record forms to complete. Food items on records were analyzed using a nutritional analysis system (Food Processor, ESHA, Salem, OR, USA) to determine total macronutrient (not reported herein) and subsequent EI per day (kcal/day).

Instructions for quantifying exercise training sessions were included along with the forms to record on. Relative to exercise sessions, assessment variables captured included: Exercise mode (i.e., running, cycling, etc.), duration (minutes per day), and intensity of daily exercise (ratings of perceived exertion (RPE); guidance was provided on how to use the RPE scale), heart rate (applicable to those participants that used heart rate monitors) and type of training session (e.g., long run, intervals, etc.). Exercise energy expenditure (EEE) was calculated using the Compendium of Physical Activity [16], using the procedures as described by Heikura and associates [9].

Participants were encouraged to contact the PI by email or in person if they had questions about how to provide proper diet/training information. Furthermore, once submitted, all diet/training records were reviewed by the PI and if any anomalies were noted, the participant was contacted to clarify issues.

### 2.4. Energy Availability (EA)

Information collected from the diet and exercise training records was utilized to estimate measures of EA. Energy availability was calculated as [4,9]:*EA = (energy intake (EI; kcals) − (exercise energy expenditure (EEE; kcals) − resting metabolic rate (RMR]/min of exercise))/kilograms of estimated**lean body mass (eLBM)*.(1)

Physical characteristics from the survey (age, height, mass) provided the details used to calculate eLBM [17] and estimate resting metabolic rate (eRMR) [18]. Energy availability status was categorized by risk level for low EA: AR: ≤30 kcal/kg eLBM; MR: 30–45 kcal/kg eLBM; and NR: ≥45 kcal/kg eLBM [6]. Risk levels were based on the research-based cut-points identified in females, as risk thresholds are currently unidentified in males. The Boer formula [17,19] incorporating height and weight (mass) was implemented to estimate LBM (eLBM). Estimated RMR was calculated using the Cunningham equation, recognized as the most appropriate for endurance athletes [20,21]. Additionally, eRMR per minute of exercise was subtracted from the EEE in the equation, as the resting caloric cost would have occurred regardless of exercise [9].

### 2.5. Statistical Analysis

Only fully completed, questionnaires were included in this analysis. Prevalence, the primary outcome of the study, was determined by the percentage of participants in each EA risk category. Additionally, ANOVA and, where appropriate, t-tests were conducted as secondary analysis to investigate mean (±SD) differences in physical and training characteristics, modes of exercise training, and effect of injury-induced exercise training breaks between EA risk groups (i.e., AR, MR, NR) (SPSS version 21, Chicago, IL). If significant F-ratios were detected in the ANOVAs, Tukey’s post hoc procedures were utilized to determine specific mean differences. Alpha level for statistical significance was set a priori at ≤0.05.

## 3. Results

### 3.1. Physical Characteristics, Energy Availability, and Prevalence

In total, 219 individuals completed the survey; however, only 108 participants completed all aspects of the study (i.e., survey, diet, and training records). Chi-square analyses were conducted to determine whether any variable might explain why some individuals did not return all study components. There were no meaningful differences to explain the lack of completion, and these data have not been reported.

A summary of the physical and training characteristics for participants is presented in Table 1 (all data are presented as mean ± SD unless otherwise indicated). The physical characteristics of the groups (see following below for risk category breakdown) were remarkably similar in age, mass, and height (no measures were significantly different from one another). The lone exception was BMI, which was significantly lower in the NR group (22.4 ± 2.3) compared to the AR group (23.7 ± 2.3, *F*_2.105_ = 3.181, *p* = 0.023).

As noted, prevalence for this study is categorized by EA risk status: (1) AR: EA ≤ 30 kcal/kg eLBM; (2) MR: EA 30—45 kcal/kg eLBM; and (3) NR: EA ≥ 45 kcal/kg eLBM. Based upon these criteria, within this sample of competitive, recreationally trained male endurance athletes, 47.2% (*n* = 51; 95% CI (37.5, 57.1)) were classified AR, 33.3% (*n* = 36; 95% CI (24.6, 43.1)) as MR, and 19.4% (*n* = 21; 95% CI (12.5, 28.2)) as NR for low EA. Actual mean (± standard deviation (SD)) EA values (kcals/kg eLBM) are shown in Table 2 for the entire sample and for the identified risk category groupings. The EA values for the risk categories were significantly different among the groupings (*F*_2.105_ = 152.443, *p* < 0.001; all differing from one another).

Energy availability and its specific components, both measured (energy intake (EI); exercise energy expenditure (EEE)) and estimated (lean body mass (eLBM) and resting metabolic rate (eRMR)) for the total sample can be found in Table 2. 

Energy availability risk status differed significantly by primary training mode (*F*_2.105_ = 4.089, *p* = 0.019). Post hoc analysis indicated that cyclists demonstrated significantly lower EA (26.9 ± 17.4 kcal/kg eLBM, *n* = 45) compared to runners (34.6 ± 13.3 kcal/kg eLBM, *n* = 55, *p* = 0.016) and the others category (39.5 ± 19.1 kcal/kg eLBM, *n* = 8, *p* = 0.037).

### 3.2. Exercise Training and Energy Availability

Relative to exercise training, the risk groups did not differ for hours per week and years of training (*p* > 0.05; see Table 1). Injury impact on training was highly variable, as 33 participants had experienced a break in their training of at least three weeks due to injury during the last 12 months (see Table 3). The prevalence of missed training due to injury tracked slightly higher in the MR (33.3%) and NR groups (47.6%), and lower in the AR group (21.5%) compared to the sample as a whole (30.6%). Subsequent analysis of how injury impacted prevalence categorization was conducted. From the entire sample, those having experienced an injury-induced training break in the previous 12 months had significantly higher EA (37.0 ± 15.7 kcal/kg eLBM) than those without a training break (29.4 ± 15.7, *t*_106_ = −2.306, *p* = 0.023). However, and most importantly, within each specific risk category, the EA was unaffected by the injury-induced training breaks (*t*_49_ = 0.523, *p* = 0.603).

### 3.3. Nutritional Supplements and Energy Availability

At least one nutritional supplement was consumed by 43.5% of participants (*n* = 47). Categories of supplements consumed are shown in Table 4 [22]. Notably, the total number of supplements consumed was greater than 47, as numerous participants consumed multiple supplements. Statistical analysis indicated supplement consumption did not significantly affect EA values across the groups (*F*_1.106_ = 1.586, *p* = 0.211).

## 4. Discussion

This study was designed utilizing an epidemiological approach to identify the prevalence of risk, as defined by the evidence-based female cut-points, for low EA within a sample of competitive, recreationally trained male endurance athletes. In addition to calculating EA, exercise mode, dietary supplementation, and injury-induced breaks to exercise training were considered as moderating variables and examined. The primary finding of this study was that only 19.4% of participants were optimizing their EA and fell into the NR category. Between the AR (47.2%) and MR (33.3%) groups, approximately 80% of the participants in this study were at some level of risk for low EA. This finding implies that low EA may have a greater prevalence among male competitive, recreational endurance athletes than previously considered. That is, low EA is not restricted only to elite male athletes engaged in exercise training.

### 4.1. Comparative Research

The majority of EA research has been conducted in female populations. As RED-S gains attention in the scientific community, more research is now addressing male athletes and the need to determine the severity of this issue and how best to prevent health and performance detriments. However, the number of male-based studies is limited, and, for that reason, findings from both sexes are addressed here.

The current study’s findings are in line with the first study of prevalence for low EA in female recreational athletes, conducted by Slater et al. [23]. These investigators found that 45.0% (*n* = 49) of participants were categorized as being at risk of low EA. It is important to point out some differences between this and the current study. The female participants were not competitively training, with eligibility requiring they meet the American College of Sports Medicine exercise guidelines [24], classifying them as exercisers instead of athletes [14]. The EA risk status was determined from completion of the ‘Low Energy Availability in Females Questionnaire’ (LEAF-Q), a screening tool for females at risk for low EA [25], instead of measured EA. Questionnaire-based screening tools have become more common recently [9,26,27], given the difficulty and lack of standardization for measuring EA [28]. Heikura et al. [9] compared questionnaire- and hormonal-level-based risk classification to measured EA status in elite male and female endurance athletes. Their findings suggested that hormonal-level-based risk classification may be more sensitive than EA status for identifying risk; however, the sensitivity appeared more appropriate in the female than the male athletes.

In a study comparing elite endurance females with and without menstrual dysfunction, both groups identified a majority of athletes with EA < 30 kcal/kg LBM, 67% and 56%, respectively [26]. Similarly, a small study (*n* = 10) in female runners and triathletes found that all of them had EA < 30 kcal/kg LBM, regardless of eumenorrheic (29 ± 4.4 kcal/kg LBM) or amenorrheic (18 ± 6.6 kcal/kg LBM) status [29]. Elevated levels of risk in female athletes is relatively common [4]. The assumption is that females experience higher levels of risk for low EA than males [6], but the current study demonstrated prevalences similar to studies in females. This is not to say they are equal, but that the risk for male athletes may be comparable; what low EA could mean for men needs further research.

While this was one of the first studies to focus exclusively on a male population (and using recreational athletes) and prevalence of risk for low EA, several studies have included both females and males. For example, Viner et al. studied cyclists (US Pro, Cat 1–4; men = 6, women = 4) who had their EA measured during pre-season, competition, and off-season. During each of the time points, the prevalence of low EA was never below 70% and, interestingly, was highest, at 90%, during the off-season [7]. No differences were noted between the sexes in measured outcomes. These authors proposed that the low carbohydrate content of the athletes’ diets was the primary stimulus for their reduced EA. However, the influence of macro-nutrient content was difficult to access in this study due to the small sample size participating. In a younger cohort (age = 16.2 ± 2.7 y) of 352 athletes, 55.7% of the male athletes (*n* = 167) had EA < 30 kcal/kg LBM. When separated by sport, the male (*n* = 22) and female (*n* = 18) endurance athletes had average EA levels of 26.9 ± 11.4 kcal/kg LBM and 36.2 ± 14 kcal/kg LBM, respectively. An investigation into the nutritional behaviors of professional jockeys, a unique population, identified an EA of 0.8 ± 12 kcal/kg LBM on race days. With an average of at least three race days per week, the authors thought it is unlikely the jockeys could make up the caloric deficit on rest days to remain in a balanced state, but rest day EA was not measured [11]. Collectively, these studies, spanning a variety of ages, sports, and sexes indicate a prevalence greater than 50% of athletes that are at risk for low EA, which agrees with the current study.

However, not all studies have agreed with such a high prevalence of low EA occurrence. Hoch et al. [30] found combined moderate and at risk EA levels (<45 kcal/kg FFM (fat free mass)) in 36% of various high school athletes and, interestingly, 39% in sedentary controls. Heikura et al. [9], implementing both measured EA and separation by low testosterone in males (25% and 41.7%) and menstrual status in females (31.4% and 37.1%), found lower risk levels as well.

The differences in EA risk prevalence might be explained by several factors. For example, the prevalence for low EA could be dependent on age, sex, sport category, level of expertise (e.g., elite to recreational), and perhaps the measurement of low EA. Unfortunately, the lowest prevalence in the reviewed studies was still 25% of participants. That is, at least one in four athletes is at some level of risk for low EA. This finding indicates a need to educate athletes about energy availability and the importance of maintaining adequate levels of food intake.

### 4.2. Are the Female Energy Availability Cut-Points Appropriate for Men?

To remain consistent with the majority of current literature, the recognized female cut-points of <30 kcal/kg LBM, 30–45 kcal/kg LBM, and ≥45 kcal/kg LBM were utilized in this study. In doing this, we were aware that there is a lack of literature confirming the use of the female cut-points as appropriate in male populations. However, to our knowledge, no literature has identified appropriate EA level cut-points that should be utilized in males.

Some researchers have suggested that cut-points should be lower in men due to reduced energy demand in the reproductive systems of males versus females [31]. Still, the female ranges have been utilized in most published male-based research except for two studies. Koehler et al. [32] implemented an optimal level of 40 kcal/kg FFM, suggesting males would see no dysfunction with a slightly lower caloric target (N.B., these researchers observed changes at <15 kcal/kg/d FFM/day). Fagerberg [33] suggested a more severe and lower EA threshold of 20–25 kcal/kg FFM in wrestlers (male) than the female-based <30 kcal/kg/d; however, wrestlers are unique athletes as they can be prone to relatively extreme bodyweight-altering tactics in their lifestyle choices. While it seems reasonable that a greater disruption of EA may be necessary in males to place men at risk, to date, there is a lack of evidence identifying the exact magnitude of decrease or clear cut-points appropriate for assessment in males. Future research must pursue the identification of appropriate cut-points for male populations.

### 4.3. Moderating Factors and Limitations

Factors such as nutritional supplementation use breaks in training due to injury and mode of training all have the potential to moderate the current prevalence findings. This was not the case, however, which suggests that long-term exercise training and dietary behavior were driving factors in the energy availability of the participants. These lacks of effect also support the notion that, even though the duration of observation of the participants was relatively short in this study, the findings seem reflective of their real-life practices.

As a prevalence study, the present sample size was relatively small. However, the demands of 3 day diet and training records in addition to the online survey likely caused substantial participant burden. Comparable prevalence size (*n* = 109) can be been found in the female EA literature [23] and in a number of prevalence studies with smaller sample sizes exist [7,9,28,29,34]. Another limitation to this study was the self-reported nature of all data, and there remains no standard for determining EA. The former is an accepted factor in designs of this approach, and the latter is an issue slowly being addressed by the research community.

The quality of the components for calculation of EA are dependent on the accuracy of the reporting from participants and the precision in the questionnaire instruments [19]. Capling et al. [35] identified shortcomings in diet records and the CHAMPs-like questionnaire formats, and the Compendium of Physical Activity which we used may not be as accurate for our aged endurance-trained athletes as for the general population. Physical activity records do, however, typically contain less misreported information than diet records [19]. Nonetheless, we acknowledge a 3 day diet record could be a factor compromising aspects of the validity of our nutritional assessment (i.e., EA calculation), as has been cautioned by some researchers who have advocated that longer assessment periods are needed [28,35]. Notably, though, some researchers have also reported good agreement between 3 day records and records collected for longer periods of time [36].

Additionally, our LBM and RMR components had to be estimated, since this study’s design was a questionnaire survey approach. In doing so, we used the Cunningham equation for RMR, which is recognized as being appropriate for endurance athletes [20,21]. We did use the Boer calculation for LBM and we acknowledge this method has limitations; however, El-kateb et al. reported that this calculation method tends to be more accurate in normal-weight adults, such as were utilized herein [37].

Finally, to our surprise, we observed a slight BMI differences between the groups (NR < AR). We feel, however, since the actual physical characteristics from which BMI is calculated were not significantly different in and of themselves, this points to the BMI differences perhaps being a statistical anomaly. Furthermore, in the energy calculations body weight (mass) was utilized and not BMI; as such we did not consider this occurrence of the BMI difference to be a critical issue.

We recognize these prior points are limitations in our data; however, this project was designed as a medical epidemiology study employing a survey questionnaire. Our approach is commonly used by practitioners in the field (e.g., registered dietitians, public health professionals), and this real-world pragmatic aspect to our study is a strength that increases the utility of our findings (i.e., increased external validity). Nonetheless, all questionnaires used in research have issues of validity and precision in the measurements, and our study was no different. We encourage researchers who pursue this topic moving forward to incorporate laboratory-based assessments to more accurately address parameters and eliminate some of these limitations, and in so doing, corroborate or refute our findings.

Finally, it should be noted that the participants in this study were competitive, recreational athletes, not elite, national-level athletes. While they were competitive in their sport, it was not their livelihood, but rather an important aspect of their lives. The generalizability of this sample, therefore, is quite broad when compared to the public-at-large, as the majority of individuals exercising are at a recreational level [12] and are not professionals ([13]; e.g., according to the U.S. Department of Labor, 21.4% of men in the USA exercise on a regular basis). Thus, the prevalence of low EA could be a relatively broad scope public health issue that most healthcare practitioners may be completely unaware of [12].

## 5. Conclusions

Participants in this study had a relatively high prevalence of being at risk for low EA. Nearly one half (47.2%) of the participants were in the AR group. Only 19.4% of participants were found to have no risk of displaying low EA. This is an alarming finding, as it indicates 80% of participants were at some degree of risk of experiencing low EA. Our prevalence of participants with low EA (<30 kcal/kg eLBM) was in agreement with much of the literature available in women and men, although the total number of available studies on men is spare. Cyclists, in particular, were at greater risk in this cohort of low EA; why this occurred is unclear and in need of further investigation. Finally, we do note that the methodological limitations of our study design and approach necessitate that caution be used in the interpretation and translation of our results to practical application.

Future research is also needed to address whether the current female cut-points are appropriate for use in male populations. Additionally, educational programs informing women and men participating in competitive, recreational endurance activities about the potential risks and warning signs of low EA are important and need to be implemented.

## Figures and Tables

**Table 1 medicina-55-00665-t001:** Summary of characteristics by energy availability risk classification status.

Characteristic	Total (*n* = 108)	At Risk (*n* = 51)	Moderate Risk (*n* = 36)	No Risk (*n* = 21)
Age (y)	38.6 ± 13.8	40.1 ± 14.6	38.0 ± 13.1	36.1 ± 13.5
Mass (kg)	74.9 ± 8.6	76.9 ± 8.2	73.3 ± 7.7	72.7 ± 10.4
Height (m)	1.80 ± 0.10	1.80 ± 0.05	1.79 ± 0.07	1.80 ± 0.09
BMI (kg/m^2^)	23.1 ± 2.3	23.7 ± 2.3	22.8 ± 2.0	22.4 ± 2.3
Exercise per week (h)	12.2 ± 5.4	12.0 ± 3.6	12.0 ± 4.1	13.0 ± 9.8
Training years at current level (y)	6.9 ± 8.7	6.5 ± 9.4	8.3 ± 9.5	5.3 ± 4.9
Training break due to injury (last 12 months) ^	33 (30.6%)	11 (21.6%)	12 (33.3%)	10 (47.6%)

EA: Energy availability. No Risk: EA ≥ 45; Moderate Risk: EA 30—45; At Risk: EA ≤ 30. ^ number of participants (percent); * significantly lower than At Risk group.

**Table 2 medicina-55-00665-t002:** Energy availability calculation variables and components by EA risk status groupings (*n* = 108) (mean ± SD).

Variables	Total *n* = 108	At Risk *n* = 51	Moderate Risk *n* = 36	No Risk *n* = 21
Energy Availability (EA) (kcal/kg eLBM)	31.7 ± 16.0	16.2 ± 10.6	34.8 ± 4.1	51.8 ± 7.8
Energy Intake (EI) (kcals/day)	3086.7 ± 810.1	2661.5 ± 708.9	3134.5 ± 474.3	4037.7 ± 667.9
Exercise Energy Expenditure (EEE) (kcals/day)	1356.4 ± 671.2	1676.9 ± 756.8	1103.3 ± 443.1	1011.9 ± 364.4
Est. Resting Metabolic Rate (eRMR) kcals/day) ^1^	1804.4 ± 102.3	1823.9 ± 88.7	1786.8 ± 96.8	1787.4 ± 134.5
Est. Lean Body Mass (eLBM) (kg) ^2^	59.3 ± 4.6	60.2 ± 4.0	58.5 ± 4.4	58.5 ± 6.1

^1^ Cunningham equation, ^2^ Boer calculation.

**Table 3 medicina-55-00665-t003:** Duration of missed training due to injury in the last 12 months by EA risk status (*n* = 33).

EA Group	*n*	≥ 3 wks–<5 wks	≥ 5 wks–<9 wks	≥ 9 wks–<12 wks	≥12 wks	Totals (%)
At Risk ^1^	51	7	2	1	1	11 (21.5)
Moderate Risk ^1^	36	3	4	0	5	12 (33/3)
No Risk ^1^	21	7	2	0	1	10 (47.6)
Total ^1^	108	17 (15.7)	8 (7.4)	1 (0.9)	7 (6.5)	33 (30.5)

^1^ Column values represents number of participants while ( ) numbers represent %, wks = weeks.

**Table 4 medicina-55-00665-t004:** Nutritional supplements consumed by participants (*n* = 47).

Supplement Category	Number of Participants Consuming
Vitamins	33
Minerals	31
Fish Oil/Flaxseed Oil	16
Protein/Amino Acids	31
Herbs/Botanicals/Extracts	14
Glucosamine Chondroitin	5
Enzymes	3
Other	3

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
