# Peer review of "Prevalence of Low Energy Availability in Competitively Trained Male Endurance Athletes"

_medicina, 2019, doi:10.3390/medicina55100665_

Round 1

Reviewer 1 Report

p.p1 {margin: 0.0px 0.0px 0.0px 0.0px; font: 12.0px Helvetica; min-height: 14.0px} p.p2 {margin: 0.0px 0.0px 0.0px 0.0px; font: 12.0px Helvetica}

Overall I think this is a very good piece of work and an important contribution to the field of endurance sports, particularly due to its focus on the recreational yet competitive athlete. Just a few small comments/suggestions/questions....

42 - would be nice to cite primary research showing increased injury risk and / or decreased athletic performance (rather than just a consensus statement)

152 - “Exploratory statistics of ANOVA and t-tests…” - can you be a bit more specific with what you did? (post-hoc testing, corrections, etc)

166 - 95% CI should be reported as (n=51; 95% CI [37.5, 57.1]) 

176 - table 1 - it’s not clear what the (*) on the energy availability variable refers to?

191 - “*significantly lower than EA≤30.” it may make more sense to say *significantly lower than At Risk, as that’s what the column is labeled as

195 - trended higher - what were the trends?

201 - do you mean p = 0.603?

204 - “data represents number or number (%), wks = weeks” —> this is confusing

290 - Some discussion or acknowledgement of the limitations of only a 3-day diet and exercise record may be warranted as well. 

For example, this study showed that it requires between 14-84 days to estimate true average intake for an individual with given statistical confidence.

https://academic.oup.com/jn/article-abstract/117/9/1638/4768567?redirectedFrom=fulltext

I realize that the food needs to be tied to the exercise, and that additional days would be a big burden on the participants. However, it is possible there are off-days where the people in at-risk categories would be moved into the no-risk, etc.

Author Response

Reviewer 1

Reviewer General Comment:

Overall I think this is a very good piece of work and an important contribution to the field of endurance sports, particularly due to its focus on the recreational yet competitive athlete. Just a few small comments/suggestions/questions....

Authors Response: Thank you for the positive comment and feedback about our manuscript. We appreciate your insights and help in making our work better.

Reviewer Specific Comments:

42 - would be nice to cite primary research showing increased injury risk and / or decreased athletic performance (rather than just a consensus statement)

Author Response: We have changed the references listed here somewhat, but the consensus statements are derived to give medical guidance to practitioners so the listing of health/performance injury/issues are very complete and more encompassing that any one individual study.

152 - “Exploratory statistics of ANOVA and t-tests…” - can you be a bit more specific with what you did? (post-hoc testing, corrections, etc)

Author Response: A change has been made as requested to clarify this section.

166 - 95% CI should be reported as (n=51; 95% CI [37.5, 57.1])

Author Response: A change has been made as requested.

176 - table 1 - it’s not clear what the (*) on the energy availability variable refers to?

Author Response: A change has been made as requested. This was an error on our part.

191 - “*significantly lower than EA≤30.” it may make more sense to say *significantly lower than At Risk, as that’s what the column is labeled as

Author Response: A change has been made as requested to clarify this explanation.

195 - trended higher - what were the trends?

Author Response: A change has been made as this was poor wording on our part, as these were not statistical trends.

201 - do you mean p = 0.603?

Author Response: Yes, you are correct, a change has been made as requested.

204 - “data represents number or number (%), wks = weeks” —> this is confusing

Author Response: A change has been made as requested to make this less confusing.

290 - Some discussion or acknowledgement of the limitations of only a 3-day diet and exercise record may be warranted as well.  For example, this study showed that it requires between 14-84 days to estimate true average intake for an individual with given statistical confidence.

https://academic.oup.com/jn/article-abstract/117/9/1638/4768567?redirectedFrom=fulltext

I realize that the food needs to be tied to the exercise, and that additional days would be a big burden on the participants. However, it is possible there are off-days where the people in at-risk categories would be moved into the no-risk, etc.

Author Response: A change has been made to point out this is as a limitation which we now draw to the reader’s attention in the Limitations section of the Discussion.

Reviewer 2 Report

The introduction does not report a huge relevance of the study, and this must be clearly presented. Moreover, some points should be clarified, in order to a reader better understood. For example, what “low energy availability” means and why is it important for health and performance? What is more important in terms of risk/health: to understand the relative energy in sport or investigating just the low energy availability?

The purpose of the study is not clear at all, given that the results do not make sense with the purpose. Authors pointed out that the study aims to investigate the prevalence of low EA; however, results present information other than not only prevalence and, moreover, the discussion talks about other aspects different from those related to the purpose.

Why studying male athletes? Further, recreational athletes can provide the same answers that those from elite athletes?

Triathlon involves cycling and running, so what is the relevance of sampled cyclist and runners, as well as triathlon athletes? Why not studying only these last ones? More, given EA seems to be related to sports where a lean body shows to be relevant, or even an aesthetic aspect is required, did authors consider sampling athletes from other sports, such as gymnastic, ice skaters, dancers?

Regarding information related to training and competition, why was this information obtained? Did authors consider the possibility to take into account information related to athletes performance (such as pace or time to conclude a competition) to evaluate differences in EA by groups?

The statistical analysis should be better described, all procedures should be mentioned, and more details must be provided. More, authors reported that they investigated the effect of injury on training breaks – does the study design allow it or just to determine “associations”?

The analysis should be performed by the different sports modalities. Did authors consider this possibility?

Athletes classified as “at risk” are those with higher exercise energy expenditure, estimated resting metabolic rate and estimated lean body mass. How this information can be explained? Does it make sense?

Discussion should be rethinking according to study purpose.

Author Response

Reviewer 2

Reviewer General Comment:

The introduction does not report a huge relevance of the study, and this must be clearly presented. Moreover, some points should be clarified, in order to a reader better understood. For example, what “low energy availability” means and why is it important for health and performance? What is more important in terms of risk/health: to understand the relative energy in sport or investigating just the low energy availability?

Authors Response: We thank the reviewer for their comment and insight on the manuscript, specifically herein the Introduction section. You will see we have altered wording and removed text to address points raised, we hope, some of your concerns are alleviated and it is clearer now. However, we will point out that this manuscript was reviewed by 4 reviewers and the other 3 reviewers did not find this part of the manuscript needed extensive work and hence we are revising with the major of the reviewers input in mind.

Reviewer General Comment:

The purpose of the study is not clear at all, given that the results do not make sense with the purpose. Authors pointed out that the study aims to investigate the prevalence of low EA; however, results present information other than not only prevalence and, moreover, the discussion talks about other aspects different from those related to the purpose.

Authors Reponses: Thank you for the comment, some re-wording has taken place as well as text removed and we hope it helps the reviewers understanding now. Please recognize that prevalence was out principle outcome and hence our focus; but, some of the addition information is provided to give a context of our participants from a demographic standpoint as well as for validation of some of our outcome interpretations. Again, however, we will point out that this manuscript was reviewed by 4 reviewers and the other 3 reviewers did not find this part of the manuscript needed extensive work and hence we are revising with the major of the reviewers input in mind.

Reviewer General Comment:

Why studying male athletes? Further, recreational athletes can provide the same answers that those from elite athletes?

Authors Response: The reason for the study of males, as stated in the manuscript is the International Olympic Committee (IOC) Medical Commission stating and noted more work is necessary in males on this topic. We chose recreational athletes because we feel there is a need for more practical application in this line of work to see if such problems occurs in the everyday person (man) who is involved with regular exercise and not just the ultra-elite athletes. Again, something recommend by experts who are working in this field of study.

Reviewer General Comment:

Triathlon involves cycling and running, so what is the relevance of sampled cyclist and runners, as well as triathlon athletes? Why not studying only these last ones? More, given EA seems to be related to sports where a lean body shows to be relevant, or even an aesthetic aspect is required, did authors consider sampling athletes from other sports, such as gymnastic, ice skaters, dancers?

Authors Responses: The prevalence of female related problems were first identified in runners, cyclist and triathlon athletes. We are modeling our work on that done in females. To move to these other sporting groups is of interest to us. But, in the context of the everyday person (man) who exercises you find that the sports we addressed are much more common (i.e., within the USA) and have far more participants in them than the sports you mention.

Reviewer General Comment:

Regarding information related to training and competition, why was this information obtained? Did authors consider the possibility to take into account information related to athletes performance (such as pace or time to conclude a competition) to evaluate differences in EA by groups?

Authors Responses: We used a medical epidemiology approach to doing this study; and in that vein chose to execute an online survey to facilitate easy of subject participating. That said, it is well recognized that subject burden (see reference 15) can become excessive with too many questions and as a result people chose not to participate because of the time committee. We attempted to balance the number of questions and information sought after to improve subject participation and hence chose not to address the points you mention in your comment. We will in our follow-up work, which is more laboratory based, be addressing this very good point you raise.

Reviewer General Comment:

The statistical analysis should be better described, all procedures should be mentioned, and more details must be provided. More, authors reported that they investigated the effect of injury on training breaks – does the study design allow it or just to determine “associations”?

Authors Response: Thank you to the reviewer for pointing this out, we have edited and changed this section to be clearer on the matter. Our epidemiological statistical expert (author KK) advised against any association analysis on your points as the number of subject participants falling into the injury groups is too small to have meaning or appropriately significant findings.

Reviewer General Comment:

The analysis should be performed by the different sports modalities. Did authors consider this possibility?

Authors Response: Again, Our epidemiological statistical expert (author KK) advised against such an analysis as the number of subject participants falling into various sports was too small to have meaning or appropriate significant findings.

Reviewer General Comment:

Athletes classified as “at risk” are those with higher exercise energy expenditure, estimated resting metabolic rate and estimated lean body mass. How this information can be explained? Does it make sense?

Authors Response: These athletes may be exercising more, and from such points would have a great stimulus for exercise training adaptations which could evoke greater muscle mass development and hence lean body mass – greater lean body mass should lead to greater resting metabolic rate. Also, keep in mind some aspects of what you point out are not statistically significant; hence, we cannot say they are different and should not address them as such.

Reviewer General Comment:

Discussion should be rethinking according to study purpose.

Authors Response: Thank you for the comments. Please note the Discussion section has been altered and revised, we hope you find it clearer and the edits have proven helpful. But, again we will point out that this manuscript was reviewed by 4 reviewers and the other 3 reviewers did not find this part of the manuscript needed a complete and highly extensive re-working and hence we are revising with the major of the reviewers input in mind.

Reviewer 3 Report

Title: Prevalence of Low energy Availability in Competitively Trained Male Endurance Athletes

Journal: Medicina

Manuscript ID: 535148

Comments to authors

General Comments:

This is an interesting and understudied area of research. The manuscript has some strengths, but also suffers from some weaknesses.  The strengths of the manuscript are mainly related to the relatively high number of respondents and that there are few studies published to date which gives this present study a value to the research field. 

The weaknesses are mainly related to the self-reported nature of all data. Furthermore, methods for conducting and interpreting energy availability (EA) is very much discussed. Therefore, the authors should be very careful when interpreting the data, especially since they have no other (objectively) markers for RED-S.

Thus, the overall drawback of this study is the method used for assessing energy availability. I find that the authors do not address and discuss these weaknesses in a satisfactory way. For instance, I recommend the authors to address in more detail the bias with self-reported energy intake (the possibility of under-reporting), subjective assessed training intensity and the use of MET-values for calculating EEE in their discussion. However, I am aware that assessing EA in athletes is extremely time consuming and we need more literature within the field of RED-S – especially with male athletes. The method used in the present study may be suitable for practitioners, which may be worth addressing.

Furthermore, I am concerned with the lack of ethical approval. Please provide detailed information on why the study was deemed exempt from review. Research in EU has strict policies of data management and storage of participants contact information, and I do not find any information regarding approval.

I have in the following some specific comments the authors may take into consideration.

Specific comments:

Introduction:

Your introduction is overall well written and allows the reader to understand your rationale and why the study is important to conduct. However, certain sentences are long and therefore difficult to read and understand.

For instance line 49-51 [Additionally though, exercise energy expenditure (EEE)…….] as well as line 61-65 could be incorporated earlier in the introduction [Most individuals participating in sport…..]

Line 66-70: Is a methodological description, not belonging in the introduction.

Materials and Methods:

Line 73; [This study was deemed from…..]. Could you please provide additional information? It is unclear why you did not apply for ethical approval.

Line 87 to 88; [The current survey was adapted from the previously validated CHAMPs physical activity questionnaire with guidance from a content expert]. The CHAMP questionnaire is problematic to use in your population, since it is validated in the elderly population. You must include this consideration in your discussion and limitations.

Line 90-92; [The survey was designed to collect descriptive…]. You should include more detailed information on how and what was used to collect and assess the descriptive data.

Line 108: [They could determine that email and then return….]” Unclear writing. Please re-phrase this.

Line 125-128; [Upon completion of the survey, participants received an automatic email providing further detail on recording three days (one weekend day and two weekdays) of dietary intake and exercise training, to measure EI and EEE, respectively. Direction for measurement and recording food was provided along with the diet record forms to complete]. This information is not adequate and need to be expanded and elucidated. It is not possible to gain an insight in how you assessed participants energy intake.

The Boer equation has its drawback, which I do not find you discuss adequately.

Line 149-155; Please add a line telling your continuous data is presented as mean +/- SD

Results:

Line 170: Finding a significant difference in EA between the EA groups is not surprising. This is probably not worth mentioning.

Please rearrange table 1 and 2. Meaning your Table 2 should be Table 1. You always present subject characteristic first. Please add P-values in the table descriptions.

Table 1: What does the * mean in the top line after eLBM?

It would be interesting to see macronutrient intake for the three groups. Could you please add carbohydrate, protein and fat per kg body mass in table 1 (should be table 2). This is relevant for your discussion (line 252).

Line 207: Here you describe how you have categorized the supplements and refer to reference 22. This should be moved to the method section. 

Discussion:

In general, you have a structured and well-written discussion regarding prevalence of low EA among male athletes and the problems of measuring and quantifying EA.

You find that those at risk for low EA have the highest BMI. This should be addressed in the discussion.

Line 282-285; [Koehler et al.  implemented an optimal level of 40 kcal/kg FFM, suggesting males would see no dysfunction with a slightly lower caloric target. Fagerberg has suggested a more severe low EA threshold of 20-25 kcal/kg FFM in wrestlers, who can be prone to relatively extreme weight control tactics.]. You should re-arrange your sentence to better address that Koehler saw changes at an EA <15 kcal/kg FFM/day.

You should discuss your results regarding the supplement intake. How can a high intake of supplements increase the risk of low EA? Pros and cons of taking supplements in an energy deficient state?  

Limitations: You should discuss your limitations at much greater depths and specifically focus on the CHAMPS questionnaire and assessment of energy intake (including the well-known bias of under reporting) as well as estimating fat-free mass.

Linje 248: Is this actually the first study focusing on male athletes? Please search the literature and update this statement.

Author Response

Reviewer 3

We thanks the review for their helpful comments and suggestions. You will see we have attempted to address each thoroughly as possible.

General Comments:

This is an interesting and understudied area of research. The manuscript has some strengths, but also suffers from some weaknesses.  The strengths of the manuscript are mainly related to the relatively high number of respondents and that there are few studies published to date which gives this present study a value to the research field.

The weaknesses are mainly related to the self-reported nature of all data. Furthermore, methods for conducting and interpreting energy availability (EA) is very much discussed. Therefore, the authors should be very careful when interpreting the data, especially since they have no other (objectively) markers for RED-S.

Thus, the overall drawback of this study is the method used for assessing energy availability. I find that the authors do not address and discuss these weaknesses in a satisfactory way. For instance, I recommend the authors to address in more detail the bias with self-reported energy intake (the possibility of under-reporting), subjective assessed training intensity and the use of MET-values for calculating EEE in their discussion. However, I am aware that assessing EA in athletes is extremely time consuming and we need more literature within the field of RED-S – especially with male athletes. The method used in the present study may be suitable for practitioners, which may be worth addressing.

Furthermore, I am concerned with the lack of ethical approval. Please provide detailed information on why the study was deemed exempt from review. Research in EU has strict policies of data management and storage of participants contact information, and I do not find any information regarding approval.

Author Responses:

1. General comment – “should be very careful when interpreting the data”:

Author Response: We have taken this remark to heart and attempted to be just that ‘very careful’ and tried not overstate the case of what our data implies. We have made certain this appears in the Conclusions so it is a final take away point from the study.

2. General comment – “do not address the weaknesses in a satisfactory way”:

Author Response: Thank you, we have extensively expanded our remarks in the Limitation section of the manuscript to point our weaknesses (i.e., self-report aspects) and encourage our readers to use caution in viewing the findings. In structuring the Limitations section of the manuscript we did look at previously published work for this journal and have attempted to model the amount of discussion text on limitations to match what is the level representative of what the journal has published previously.

We also do wish to emphasize to the reviewer that this study used a medical epidemiology approach to the research question and one of our authors (KK) is a highly regarded epidemiologist at our university who studies sport, sport-injury and exercise. They carefully reviewed the procedures we used in designing the study and deemed them with limitations, but appropriate and within the standard of practice for the field.

Furthermore the dietary information gathered did involve the subjects have prompting information and guidance on portion size and quantity during completing of their questionnaire in order to aid in our information accuracy. We do agree that the approaches we used and the design model does fit well with practitioners in the field and have added comments to such, thank you. We have made comment relative that in the conclusions.

3. General comments – “ethical approval”:

Author Response: Our study did undergo review by our university ethical review committee, and the subjects did have to give an informed consent, there were no ethical violations and we were in accordance with the Helsinki Declaration. However, we recognize the wording we used did not convey this clearly and we have re-written this section. We have provided documentation to support this claim to the journal.

Specific Comments:

Introduction:

Your introduction is overall well written and allows the reader to understand your rationale and why the study is important to conduct. However, certain sentences are long and therefore difficult to read and understand.

For instance line 49-51 [Additionally though, exercise energy expenditure (EEE)…….] as well as line 61-65 could be incorporated earlier in the introduction [Most individuals participating in sport…..]

Author Response: We have edited aspects of the sentences you mention, and moved some text per your request. However, we feel strong the “flow and structure” of the text topic is in appropriate order.

Line 66-70: Is a methodological description, not belonging in the introduction.

Author Response: We have edited, removed and move some aspects of these sentences per your request.

Materials and Methods:

Line 73; [This study was deemed from…..]. Could you please provide additional information? It is unclear why you did not apply for ethical approval.

Author Response: We did have ethics approval and have re-worded this section to support that claim.

Line 87 to 88; [The current survey was adapted from the previously validated CHAMPs physical activity questionnaire with guidance from a content expert]. The CHAMP questionnaire is problematic to use in your population, since it is validated in the elderly population. You must include this consideration in your discussion and limitations.

Author Response: Thank you for pointing out this concern, we have made note of it and commented in the Limitations of the manuscript concerning this point. Our choice was based upon the assumption that most of our participants would be older adults who had been engaged in exercise activities for a number of years. Nonetheless, we certainly see your point.

Line 90-92; [The survey was designed to collect descriptive…]. You should include more detailed information on how and what was used to collect and assess the descriptive data.

Author Response: Thank you for your recommendation, but we feel the information reported on the subjects/participant demographics in the Results points to the what was the descriptive data collected, and to include more information here would be redundant as it appears obvious in the Results. We have place some degree of remarks to change the text here but not excessively.

Line 108: [They could determine that email and then return….]” Unclear writing. Please re-phrase this.

Author Response: Edited and re-written. Thank you.

Line 125-128; [Upon completion of the survey, participants received an automatic email providing further detail on recording three days (one weekend day and two weekdays) of dietary intake and exercise training, to measure EI and EEE, respectively. Direction for measurement and recording food was provided along with the diet record forms to complete]. This information is not adequate and need to be expanded and elucidated. It is not possible to gain an insight in how you assessed participants energy intake.

Authors Response: This is describing the particular elements used in calculating the components of energy intake/output and ultimately EA. Subsequently we provide formula utilized as well as extensive references for the reader to pursue on the topic of the steps and procedures. Since this is not a “methodology paper” we want to not overly burden the reader with excessive detail that is readily available in the literature. Nonetheless, we have reworded aspects of this slightly.

The Boer equation has its drawback, which I do not find you discuss adequately.

Author Response: Thank you for point this out. We had already alluded to this point in the Limitations section of the Discussion, but we have edited our text to be clearer about this point.

Line 149-155; Please add a line telling your continuous data is presented as mean +/- SD

Author Response: Corrected per your request.

Results:

Line 170: Finding a significant difference in EA between the EA groups is not surprising. This is probably not worth mentioning.

Author Response: While note perhaps not surprising but we would like for the sake of clarity to keep this within the manuscript.

Please rearrange table 1 and 2. Meaning your Table 2 should be Table 1. You always present subject characteristic first. Please add P-values in the table descriptions.

Author Response: Thank you, but we respectfully decline. We feel it more important to show the main aspects of the study Prevalence should that different groups did exist prior to describing them. It seems illogical to us to take about descriptive difference of groups before we have actually defined/determined there are different groups to begin with.

Table 1: What does the * mean in the top line after eLBM?

Author Response: Removed, a typographic error on our part.

It would be interesting to see macronutrient intake for the three groups. Could you please add carbohydrate, protein and fat per kg body mass in table 1 (should be table 2). This is relevant for your discussion (line 252).

Author Response: This particular data will appear elsewhere in a subsequent manuscript related to this study; and, we now denote this point.

 Line 207: Here you describe how you have categorized the supplements and refer to reference 22. This should be moved to the method section.

Author Response: There is some aspects of methodology here, but extremely slight and minor. We prefer to leave this as it as we feel the flow of the text hear lends itself to it.

Discussion:

In general, you have a structured and well-written discussion regarding prevalence of low EA among male athletes and the problems of measuring and quantifying EA.

Author Response: Thank you.

You find that those at risk for low EA have the highest BMI. This should be addressed in the discussion.

Author Response: We understand your point, but the individual group height’s and body masses were not statistical significantly different and the calculated BMI was, we feel this represents a statistical anomaly and not meaningful since the variables BMI are calculated from are not significantly different. We have made remakes to this affect and not it in the Discussion of the manuscript.

Line 282-285; [Koehler et al.  implemented an optimal level of 40 kcal/kg FFM, suggesting males would see no dysfunction with a slightly lower caloric target. Fagerberg has suggested a more severe low EA threshold of 20-25 kcal/kg FFM in wrestlers, who can be prone to relatively extreme weight control tactics.]. You should re-arrange your sentence to better address that Koehler saw changes at an EA <15 kcal/kg FFM/day.

Author Response: Sentence has been edits per your request.

You should discuss your results regarding the supplement intake. How can a high intake of supplements increase the risk of low EA? Pros and cons of taking supplements in an energy deficient state? 

Author Response: With all due respect, we decline to address this issue in the Discussion as it would very quickly become long and cumbersome and we think be off topic and detract from the point of the study – “what is the prevalence of At Risk” athletes”?

Limitations: You should discuss your limitations at much greater depths and specifically focus on the CHAMPS questionnaire and assessment of energy intake (including the well-known bias of under reporting) as well as estimating fat-free mass.

Author Response: We have edit this part of the text to address this point and make note of it.

Line 248: Is this actually the first study focusing on male athletes? Please search the literature and update this statement.

Author Response: To the best of our knowledge it is the first to address prevalence in this population of male athletes, competitive but recreational in nature (i.e., not elite). We have rephrases our text to reflect that sentiment.

Reviewer 4 Report

This is a nicely presented paper. I am concerned about the methodology used, which may have affected the results obtained, the implications of which are not discussed sufficiently, if at all. Participants in the AR group were heavier and have higher BMI's than the NR group. This in itself appears a contradiction which has not been referred to within the discussion. In addition participants with the highest BMI had the lowest energy intake, pointing to the fact that perhaps it was energy intake calculation error, inaccurate food diaries completed, which may have caused a lower energy intake to be observed. However much of the basis for risk categorisation is based upon energy intake calculations, which if not absolutely correct will either underestimate or overestimate risk assessment. This has not been referred to at all within the discussion.

Was anything included within the questionnaire given to participants to get an idea of what their perception of portion size was? I would suggest this could be used as a validation method to ensure participant accuracy in portion size estimation. Without some method of ensuring participants could accurately estimate portion sizes, the results appear of little value and somewhat contradictory. Is it possible to include a measure of portion size accuracy determination to increase the validity of this research, since as the results stand it looks somewhat unrealistic, heavier recreational athletes, ate more, exercised more and so were at risk of REDS compared to lighter athletes who ate more and exercised less. This is the part which is difficult to get the reader's head around and indicates warning lights for data accuracy.

Line 197-200 - needs to be addressed in the discussion. Again it seems strange those with higher injury induced training breaks had higher BMI?

Table 2: change weight to "body mass"

Figure 1: show presence of significant differences between groups on the chart.

Author Response

Reviewer 4

Reviewer Comment:

This is a nicely presented paper. I am concerned about the methodology used, which may have affected the results obtained, the implications of which are not discussed sufficiently, if at all. Participants in the AR group were heavier and have higher BMI's than the NR group. This in itself appears a contradiction which has not been referred to within the discussion. In addition participants with the highest BMI had the lowest energy intake, pointing to the fact that perhaps it was energy intake calculation error, inaccurate food diaries completed, which may have caused a lower energy intake to be observed. However much of the basis for risk categorisation is based upon energy intake calculations, which if not absolutely correct will either underestimate or overestimate risk assessment. This has not been referred to at all within the discussion.

Authors Response:

We thank the reviewer for pointing out this issue about the BMI. It is important to recognize that the body weight/mass was not significant difference among the risk groups. Furthermore, there was no significant differences between the groups for height either. Yes, the AR group and No Risk group had differences in the calculated variable of body mass index (BMI), but we felt that the actual physical characteristics from which it is calculated not being significantly different in and of themselves only points to the BMI differences being a statistical anomaly. Furthermore, in the energy calculations weight/mass comes into play and not BMI so we feel this is not such a critical issue. We have made note of this and have offered some commentary in the revised manuscript version. Furthermore, the likelihood of participants under or over reporting is a limitation in such epidemiological survey work that exist across the field of study. We would also point out that such error would be randomly distributed across all of the subjects and hence likely to affect each resulting risk classification group equally and not targeted to one classification over another. We have added some remark to this effect, thank you.

Reviewer Comment:

Was anything included within the questionnaire given to participants to get an idea of what their perception of portion size was? I would suggest this could be used as a validation method to ensure participant accuracy in portion size estimation. Without some method of ensuring participants could accurately estimate portion sizes, the results appear of little value and somewhat contradictory. Is it possible to include a measure of portion size accuracy determination to increase the validity of this research, since as the results stand it looks somewhat unrealistic, heavier recreational athletes, ate more, exercised more and so were at risk of REDS compared to lighter athletes who ate more and exercised less. This is the part which is difficult to get the reader's head around and indicates warning lights for data accuracy.

Authors Response:

The dietary information gathered did involve the subjects have prompting information and guidance on portion size and quantity during completing of their questionnaire in order to aid in our information accuracy. This information had to be limited so as to not make the questionnaire so long and burdensome that no one completed it, but they did get guidance including examples of portion size. We have added text to the revised manuscript to reflect this point.

Reviewer Comment:

Line 197-200 - needs to be addressed in the discussion. Again it seems strange those with higher injury induced training breaks had higher BMI?

Authors Response:

Please see remarks above, but as noted comments have been added to the manuscript.

Reviewer Comment:

Table 2: change weight to "body mass"

Authors Response:

Corrected throughout the manuscript.

Reviewer Comment:

Figure 1: show presence of significant differences between groups on the chart.

Authors Response:

Corrected as requested.

Round 2

Reviewer 4 Report

Overall I do not think the changes made have actually made the manuscript better. In some ways it has made the manuscript more difficult to read at times. Phrasing while it was good in the original manuscript, not appears to have deteriorated in several places, with some of the additions.

While I have some specific comments within the manuscript, my overall feeling with regard to this manuscript is similar to my previous opinion. The method used is not appropriate to accurately determine energy intake, and hence availability. This has been referred to within the manuscript itself, line 247. I asked for additional measures to be included to confirm the accuracy of self reported data, but I do not find the changes completed are enough. Ultimately I feel to say higher BMI within an at risk energy availability group is " a statistical anamoly" is not a good enough consideration. I would not be happy to accept this article for publication

Other comments -

Line 56 - a more recent reference should be used e.g Melin 2019

Line 62 - Rephrase

Line 64 - The majority of research is at elite, professional level, how do you know - use reference to back up this statement

Line 150-151, add a comma

The results are presented in a confusing manner. Table 2 should be the first table presented to give a feel for who the participants were and their characteristics.

What is the value of adding figure 1 when these results are already presented in Table 2, line 1?

If you do post hoc analysis for different sports types and find a significant difference depending on sport, what were the physical characteristics of these participants?

Author Response

I have added this response in an attached file too:

Response to Revision of Version # 2 of Manuscript 535148 and Reviewer # 4 Remarks

Dear Journal and Reviewer # 4:

Thank you for your continued feedback to help make our manuscript better. Below are the comments provided by the reviewer which I have “broken apart into segments” so we might offer feedback and commentary on each point separately.

We hope out responses and the changes made in the manuscript are acceptable to you and the journal such that it can now be accepted for publication.

ACH for the authors

-------------------------------------------

Comments and Suggestions for Authors

Reviewer # 4 – Major Issue # 1:

Overall I do not think the changes made have actually made the manuscript better. In some ways it has made the manuscript more difficult to read at times. Phrasing while it was good in the original manuscript, not appears to have deteriorated in several places, with some of the additions.

Response –

Thank you for this observation. We had made changes based upon your prior recommendations as well as that of the other three reviewers of the manuscript. We have with the latest version (#3) of the manuscript done an extensive re-read manuscript for language and textual transition as well as had the document reviewed by the language grammatical program service of our university. We hope you find these alterations are acceptable.

Reviewer # 4 – Major Issue # 2:

While I have some specific comments within the manuscript, my overall feeling with regard to this manuscript is similar to my previous opinion. The method used is not appropriate to accurately determine energy intake, and hence availability. This has been referred to within the manuscript itself, line 247. I asked for additional measures to be included to confirm the accuracy of self reported data, but I do not find the changes completed are enough.

Response –

First and foremost I want to recognize Reviewer # 4 for their scholarship in reviewing the work and staying with their principles on what they find at fault about our study.

It is quite obvious the reviewer is concerned about our study design-approach and is not in favor of it, and wants different and additional data collected to support and, or validate our outcomes.

In the study, we did initiate steps to provide the participants with information on portion size and quantity through handouts provided and online site guidelines, and we required them to review this material. Additionally, we informed participants in the Informed Consent that if we had questions on diet issues or they had questions on training issues that we would/could break the anonymous nature of the survey and pursue dialogue (which did happen), and if they were physical located near our research headquarters they could have one on one discussions with the study PI (this also happened). We feel these steps increased the validity of aspects of the data collected. The manuscript now has commentary to this effect and we hope the reviewer feels more confident about our data.

But, we do wish to be very clear, we cannot go back and collect more or different data and this study is complete and cannot be re-opened. I am sorry the reviewer does not feel a survey approach is appropriate for this study – but we have two public health nutrition professors as well as an epidemiology professor as authors on the manuscript who helped develop the design and study approach and they feel as an epidemiological study it is constructed appropriately. Their opinions are supported in the research literature (https://www.ncbi.nlm.nih.gov/pmc/articles/PMC5325124/). Furthermore, we very explicitly state in the manuscript in several places this is an epidemiological based study and not a laboratory based study.

I will also remind the editor and the Reviewer that this manuscript was reviewed initially by four different reviewers, three of which wrote positivity about the manuscript, study design and approach. In dealing with the assistant editor of the journal, Svetlana Petric, who handled the initial review I asked about having divergent opinions of reviewers – that is, your initial review (Reviewer # 4) was relatively negative and the other three reviewers were highly positive – what was I to do in such a situations. I was advised to answer your concerns as best I could, but to rely more on the majority opinion of reviewers as to what were significant changes to be addressed in revising the manuscript. 

We note there are limitations to the approach we used, and clearly acknowledged this very openly in the prior version of the manuscript (lines 341-348), which has now been further revised to stress these points. The reviewer gives the impression this is not adequate acknowledgement. We are uncertain how we can be more clear and indicate and emphasize these limitations to the journal readers. We feel that if a reader were of the opinion these limitations are too much and they cannot accept the outcome of the data – then they can chose to discount the study and ignore it. In other words the reader of this work can make their own judgement as to whether these are severe limitations. There is nothing we can do in a post-hoc fashion to remove these limitations since the study is completed and closed from further data acquisition.

Reviewer # 4 – Major Issue # 3

Ultimately I feel to say higher BMI within an at risk energy availability group is " a statistical anamoly" is not a good enough consideration. I would not be happy to accept this article for publication

Response –

Again we point out that the factors (height, mass) from which BMI are calculated for the groups does not different; and the calculations used in study outcome parameters as based upon body weight (mass). This latter seems to us a far more critical and important issue in the context of what the study is looking at – which we state clearly in lines 339-340 of the previous version of the manuscript. We have however edited this section to explain our point of view more thoroughly.

We fail to see now this one issue causes our data to be compromised to the point it invalidates our interpretations. And, again, we leave it to the readers of this work can make their own judgement as to whether this is a major point to consider or not. We are trying to be very fore right, honest and clear in pointing such things out to the reader and not just ignore them

Also, perhaps the more critical point here is that then BMI categories among the groups does not change (https://www.cdc.gov/healthyweight/assessing/bmi/adult_bmi/index.html); that is, they are all in the “normal, healthy weight” status.

Reviewer # 4 – Minor Issues

Line 56 - a more recent reference should be used e.g Melin 2019

Response – Thank you for this suggestion, we have incorporated the reference you request.

Line 62 – Rephrase

Response – Thank you, we have edited this line per your request.

Line 64 - The majority of research is at elite, professional level, how do you know - use reference to back up this statement

Response – Thank you we have added additional references to address this point per your request.

Line 150-151, add a comma

Response - Thank you we have added additional punctuation to address this point per your request.

The results are presented in a confusing manner. Table 2 should be the first table presented to give a feel for who the participants were and their characteristics.

Response – Thank you for bringing this to our attention. We have reformatted the Results section to be more in line with what you would like to see organizationally.

What is the value of adding figure 1 when these results are already presented in Table 2, line 1?

Response – Per your request, we have removed Figure 1 and it is now no longer part of the manuscript.

If you do post hoc analysis for different sports types and find a significant difference depending on sport, what were the physical characteristics of these participants?

Response – These responses were not substantial and, or significantly different. But, the application of the statistical analysis is limited due to some sample size issues as there are only a few respondents for some sports (too few to run statistics in a valid fashion).
